# Magnon-phonon Fermi resonance in antiferromagnetic CoF$_2$

Thomas W. J. Metzger ●[1] ✉, Kirill A. Grishunin ●[1], Chris Reinhoffer[2], Roman M. Dubrovin ●[3], Atiqa Arshad[4], Igor Ilyakov[4], Thales V. A. G. de Oliveira ●[4], Alexey Ponomaryov[4], Jan-Christoph Deinert ●[4], Sergey Kovalev ●[4], Roman V. Pisarev ●[3], Mikhail I. Katsnelson ●[1], Boris A. Ivanov ●[1], Paul H. M. van Loosdrecht ●[2], Alexey V. Kimel ●[1] & Evgeny A. Mashkovich ●[2] ✉

Understanding spin-lattice interactions in antiferromagnets is a critical element of the fields of antiferromagnetic spintronics and magnonics. Recently, coherent nonlinear phonon dynamics mediated by a magnon state were discovered in an antiferromagnet. Here, we suggest that a strongly coupled two-magnon-one phonon state in this prototypical system opens a novel pathway to coherently control magnon-phonon dynamics. Utilizing intense narrow-band terahertz (THz) pulses and tunable magnetic fields up to $\mu_0 H_{ext} = 7$ T, we experimentally realize the conditions of magnon-phonon Fermi resonance in antiferromagnetic CoF$_2$. These conditions imply that both the spin and the lattice anharmonicities harvest energy from the transfer between the subsystems if the magnon eigenfrequency $f_m$ is half the frequency of the phonon $2f_m = f_{ph}$. Performing THz pump-infrared probe spectroscopy in conjunction with simulations, we explore the coupled magnon-phonon dynamics in the vicinity of the Fermi-resonance and reveal the corresponding fingerprints of nonlinear interaction facilitating energy exchange between these subsystems.

The lattice dynamics of crystals involve interdependent periodic movements of individual atoms, rendering them inherently complex. Nevertheless, the modern theory of condensed matter has successfully managed to describe the dynamics in terms of linear superposition of mutually independent phononic modes comprising acoustic and optical phonons. However, if the amplitude of the lattice vibrations is large, this approximation fails[1–4], and the lattice dynamics enter a poorly explored regime, in which phonon anharmonicity opens up new channels of energy transfer between otherwise non-interacting modes. Several other works have shown a nonlinear interaction regime for a variety of magnonic systems[5,6].

In 1931[7], Enrico Fermi reported about the interaction of seemingly non-interacting vibrational modes in carbon dioxide CO$_2$ molecules whose frequencies differ by a factor of two. It was suggested that this resonance should be accompanied by a resonant energy transfer between the modes[8–12]. The phenomenon of nonlinear coupling, satisfying the conditions set forth by Fermi, was also observed in magnon dynamics[13] resulting in shape deformations and broadening of the respective magnon spectrum, as well as in nano-mechanical systems showing coupled vibrational dynamics[14]. In general, such behavior is expected between any eigenmodes $x$ and $y$ that involve a nonlinear coupling term $x^2 y$[15]. Several recent works demonstrated nonlinear interactions between otherwise orthogonal states, for instance, two-phonon-one-magnon[16], two-magnon-one-magnon[17,18], and two-phonon-one-phonon scattering[19–23] processes. However, Fermi resonance in the strong coupling regime is distinct and can be

[1]Institute for Molecules and Materials, Radboud University, Heyendaalseweg 135, Nijmegen 6525 AJ, The Netherlands. [2]Institute of Physics II, University of Cologne, Zuelpicher Straße 77, Cologne 50937, Germany. [3]Ioffe Institute, Russian Academy of Sciences, St. Petersburg 194021, Russia. [4]Institute of Radiation Physics, Helmholtz-Zentrum Dresden-Rossendorf, Bautzner Landstraße 400, Dresden 01328, Germany. ✉e-mail: thomas.metzger@ru.nl; mashkovich@ph2.uni-koeln.de

seen as modification of electronic states. With our work, we seek to explore this strongly coupled regime of Fermi resonance.

Interestingly, the recently demonstrated nonlinear excitation of a phonon mode mediated by a magnon state suggests the presence of such a term in antiferromagnetic $CoF_2$[24]. We anticipate that in the vicinity of the resonance, two-magnon-one-phonon interaction will affect the coupled dynamics dramatically. However, in zero applied magnetic field the double magnon frequency $2f_0$ of at $T = 6\,K$ is higher than the frequency $f_{ph}$ of the $B_{1g}$ phonon $2f_0 > f_{ph}$[25] and thus the system is not in resonance (see Fig. 1a). This is why we propose to apply an external magnetic field along the magnetic easy-axis, which splits the magnon branches, while the phonon frequency remains unchanged[26]. Particularly, a field of $\mu_0 H_{ext} = 4\,T$ is expected to be sufficient to reach the frequency matching condition (Fig. 1a, black star) with a lower energy magnon branch $2f_m = f_{ph}$, where the conditions of Fermi resonance might be satisfied. In Fig. 1b, we illustrate the aforementioned processes by a graphical representation for an off-resonant system (I) and the magnon-phonon subsystem under Fermi-resonance condition (II). A pictorial representation of the energy transfer is depicted by Feynman diagrams in Fig. 1c, illustrating the magnon-phonon interaction.

Performing THz pump-infrared (IR) probe spectroscopy in combination with simulations, we reveal the corresponding fingerprints of coherent energy exchange driven by a THz stimulus in the vicinity of the Fermi-resonance. We anticipate that this opens up new opportunities to tailor the excitation, selectively driving complex magnon-phonon dynamics within the material.

## Results

$CoF_2$ belongs to the class of antiferromagnets with a rutile-type crystallographic lattice[27], described by the $P4_2/mnm$ space group. The multi-atomic primitive cell forms 11 optical phonon modes[28,29]: $A_{1g} \oplus A_{2g} \oplus A_{2u} \oplus B_{1g} \oplus 2B_{1u} \oplus B_{2g} \oplus 3E_u \oplus E_g$. The lowest-lying Raman-active phonon mode has $B_{1g}$ symmetry and is centered at a frequency of $f_{ph} = 1.96\,THz$ at $T = 6\,K$. It is worth noting that the frequency of this mode remains the same in external magnetic field up to $\mu_0 H_{ext} = 7\,T$.

The spins of $Co^{2+}$ ions are aligned along the crystallographic c-axis below the Néel temperature of $T_N = 39\,K$. In our experiment, we use a

500 µm-thick single crystal $CoF_2$ plate cut perpendicular to the c-axis. If no magnetic field is applied, there is a doubly degenerate antiferromagnetic resonance mode at the frequency $f_0 = 1.14\,THz$ (at 6 K). Applying an external magnetic field, one breaks the degeneracy of the respective magnon mode. For instance, if a magnetic field is applied along the c-axis with its value below the spin-flop field threshold of $\mu_0 H_{ext} = 14\,T$[30], the frequencies of two degenerate magnon modes obey the relation $f_m = f_0 \pm \gamma H_{ext}$, where $\gamma$ is the gyromagnetic ratio.

Using the intense, spectrally dense superradiant THz source TELBE located at Helmholtz-Zentrum Dresden-Rossendorf[31] in combination with external magnetic fields $H_{ext}$, we get the unique opportunity to pump the magnon selectively while controlling its center frequency $f_m$ by tuning $H_{ext}$ at our disposal. This configuration allows one to explore the spin-lattice interaction in the vicinity of the Fermi resonance $2f_m = f_{ph}$ by monitoring the phonon response. As it has been shown earlier[24], as long as both the magnon and the phonon maintain their coherence, they will induce transient optical anisotropy in the originally isotropic (ab) plane of the antiferromagnet and thus modulate various components of the dielectric permittivity[32], which we track by changes of the probe-polarization[24]. The experimental geometry and the THz pulse characteristics are provided in Fig. 2a. The THz field strength was estimated to be of the order of 100 kV/cm. The THz-induced rotation of the probe-polarization measured in external magnetic fields up to $\mu_0 H_{ext} = 7\,T$ is plotted in Fig. 2b. Evidently, the time domain signal exhibits clear oscillations that are significantly influenced by an external magnetic field. Here, the magnon amplitude dominates from −10 to 70 ps whereas for the range 70–120 ps, the phonon seems to be exclusively present. In Supplementary Section A, we illustrate our observations by sinusoidal fitting of the respective magnon and phonon oscillations.

Performing Fourier transformation of the whole time domain range (−10 to 120 ps) in Fig. 3a reveals the presence of the magnon response oscillating at frequency $f_m$, and a second, distant peak at the $B_{1g}$ phonon frequency $f_{ph}$. For magnetic fields of $\mu_0 H_{ext} = 0\,T$ and $\mu_0 H_{ext} = 7\,T$, the THz pump spectrum barely covers the magnon mode and THz-induced polarization rotation contains a substantial contribution of the spectrally broad forced magnetic response closely

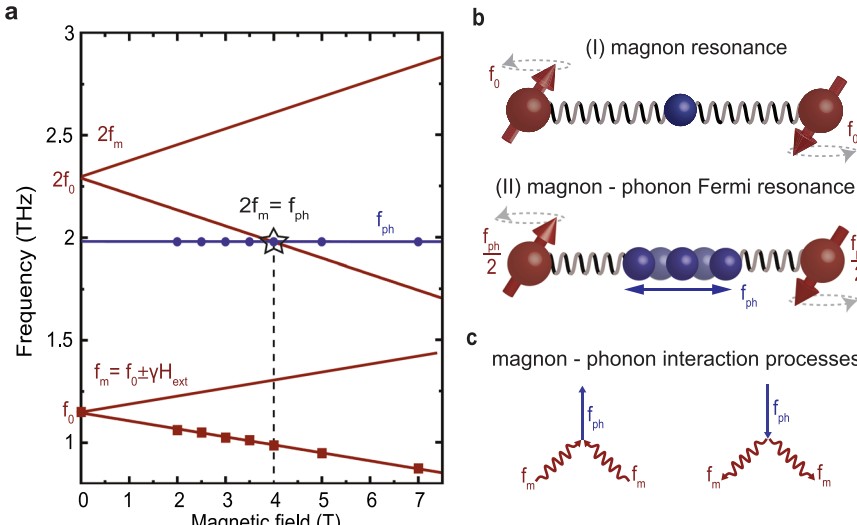

**Fig. 1 | The pathway to magnon-phonon Fermi resonance. a** Frequency tuning of the fundamental magnon frequency $f_0$ and double frequency $2f_0$ by an external magnetic field $\mu_0 H_{ext}$ applied along the antiferromagnetic easy axis of $CoF_2$. The Fermi resonance matching condition $2f_m = f_{ph}$ at the crossing of the double magnon frequency (red) and the phonon frequency (blue) is marked by a black star. Blue dots and red rectangles correspond to real experimental data points. **b** Graphical illustration of the nonlinear magnon-phonon dynamics. If the magnon frequency is

$f_0$, a THz pump pulse exclusively populates the magnon state. However, by tuning the resonance condition by an external magnetic field, for $2f_m = f_{ph}$, the magnon-phonon Fermi resonance condition is fulfilled and a channel of nonlinear energy transfer opens. **c** Feynman diagrams illustrating the processes of energy transfer involving two magnons and one phonon. Two magnon-phonon confluence (left) and phonon-two magnon splitting (right).

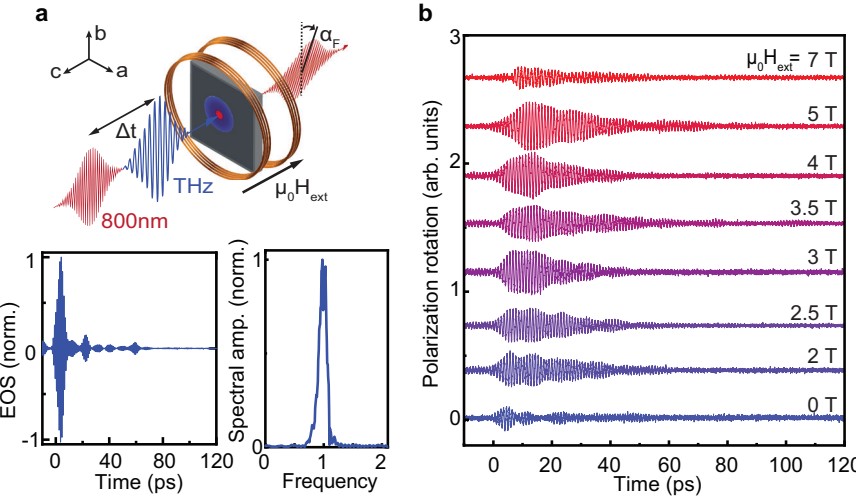

**Fig. 2 | Experimental setup and results. a** THz pump−IR probe spectroscopy with external magnetic field applied perpendicular to the sample plane, along the c-axis. The changes of THz-induced probe polarization rotation $\alpha_F$ are measured by a balanced photodetector. Electro-optical sampling of the THz pulse (with nitrogen purge) is shown in both the time and frequency domains. **b** Time domain data for polarization rotation $\alpha_F$ for a series of external magnetic field values measured at $T = 6$ K.

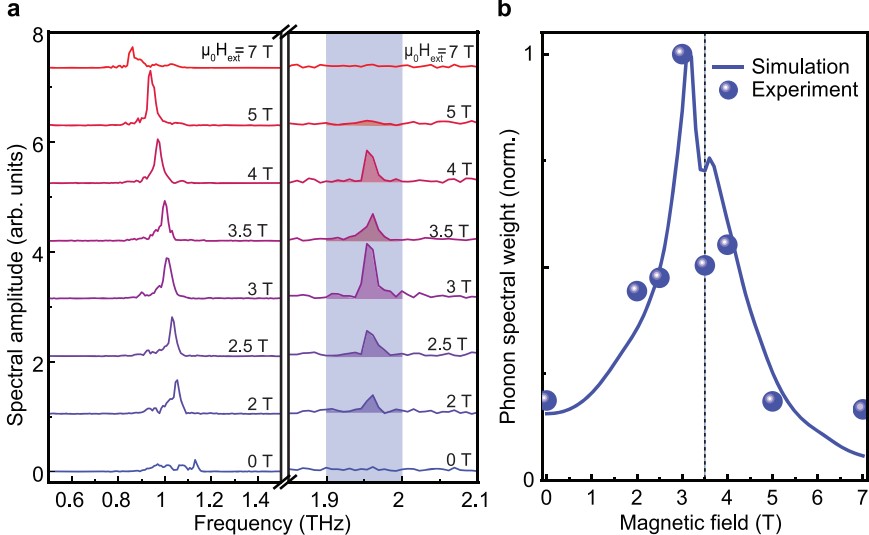

**Fig. 3 | Fingerprints of the magnon-phonon Fermi resonance. a** Fourier transformation of the time domain signals in Fig. 2b plotted with equidistant offset in the vicinity of the magnon (left) and phonon (right) resonance frequencies. To compensate for the THz power change of the TELBE source, magnon (phonon) spectral amplitudes are normalized by the square root of THz power (the THz power). **b** Phonon spectral weight (blue spheres) extracted as an integral value from the shaded frequency interval in (**a**). The phonon weight from our numerical calculation is shown as solid blue line. The vertical dotted line emphasizes the drop in the phonon spectral weight at $H_{ext} = 3.5$ T for both simulation and experiment. Excluding effects which can additionally contribute to our signal on the time scale of our THz pump pulse, i.e., nonlinear electro-optical Kerr effect, we demonstrate even better agreement of experiment and simulation in Supplementary Section A.

following the magnetic field of the THz pulse, see Fig. 2b. Moreover, no phonon-induced dynamics are observed at these fields, implying that nonlinear excitation of phonons via the mechanism described in ref. 1 does not play a significant role here. Closer to the Fermi resonance for the in-between magnetic fields of $\mu_0 H_{ext} = 2\text{--}5$ T, we observe the low energy magnon branch $f_m$ with its frequency linearly decreasing with external magnetic field. Remarkably, the strongest magnon peak at $\mu_0 H_{ext} = 5$ T does not correspond to the strongest phonon peak, revealing complex dynamics in the vicinity of the magnon-phonon Fermi resonance.

The most peculiar feature is observed at $\mu_0 H_{ext} = 3.5$ T, see Fig. 3a. Firstly, the phonon peak amplitude for the dynamics at this magnetic field is substantially reduced with respect to the peak amplitudes for $\mu_0 H_{ext} = 3$ T and 4 T. Secondly, the phonon spectrum at $\mu_0 H_{ext} = 3.5$ T

becomes broader. In fact, this resembles a splitting of the spectral line reported for purely phononic[8] or purely magnonic[13] systems under continuous wave pumping in vicinity of their Fermi resonances. To capture the energy redistribution, we integrate the area under the phonon spectra for different external magnetic fields $H_{ext}$ over the shaded range of 1.9−2.0 THz, and retrieve the behavior of the phonon resonance curve as shown in Fig. 3b. Here, the phonon resonance line is clearly asymmetric with a pronounced dip at $\mu_0 H_{ext} = 3.5$ T indicating non-trivial magnon-phonon energy exchange. In the following section, we assign this feature to the unique benchmarks of magnon-phonon Fermi resonance. Moreover, we address the magnon and phonon lifetimes and reveal the role of the nonlinear coupling constants for elevating the system into the strong nonlinear magnon-phonon coupling regime.

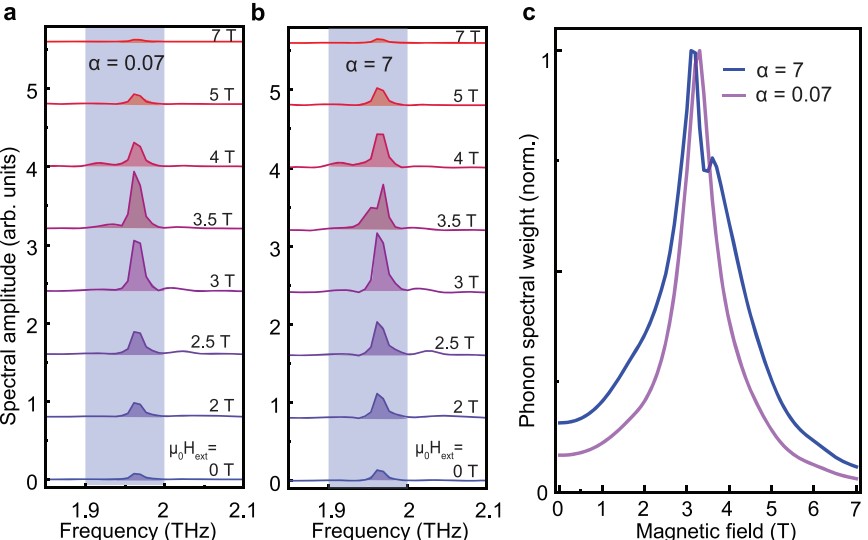

**Fig. 4 | Theoretically modeled effect of the nonlinear coupling strength on the observation of magnon-phonon Fermi resonance. a** Fourier transformation centered around the phonon resonance for the case of weak coupling with $\alpha = 0.07$. **b** Fourier transformation centered around the phonon resonance for the case of strong coupling with $\alpha = 7$. **c** Phonon weight extracted as an integral value in the shaded frequency region of 1.9–2.0 THz for the weak coupling case (purple line) and the strong coupling case (blue line), respectively. The evident difference for FFT line shapes in the range of $\mu_0 H_{ext} = 3–4$ T (**a**, **b**) and extracted phonon weight (**c**) clearly emphasizes the mutual coupling between the two magnons and the phonon state.

To obtain a better insight on the observed fingerprints of the magnon-phonon Fermi resonance, we undertook a simulation of the signatures of nonlinearly coupled dynamics in $CoF_2$. Conventionally, antiferromagnetic spins are described in terms of the Néel vector $\mathbf{L} = \mathbf{M}_1 - \mathbf{M}_2$, where the net magnetic moments $\mathbf{M}_{1,2}$ are formed by $Co^{2+}$ ions at the center and the corners of the unit cell, correspondingly[33]. The movement of the $B_{1g}$ phonon is characterized by the phonon coordinate $\theta_{ph}$. The perturbations from the ground state are introduced as $\mathbf{L}(t) = (l_x(t), l_y(t), L_0)$. Here, $L_0$ describes the ground state Néel vector. The rule of Fermi resonance symmetry implies that the $B_{1g}$ phonon symmetry $(x^2 - y^2)$ should follow the symmetry of the double magnon excitation. Hence, the corresponding nonlinear term can be introduced in the Lagrangian as $\Phi = -\alpha(l_x^2 - l_y^2)\theta_{ph}$[15], where $\alpha$ represents the nonlinear coupling constant between the magnon and the phonon subsystems. We assume that the magnetic field of the THz pulse $\mathbf{h}_{THz} = (h_x, h_y, 0)$ is polarized exclusively in the sample plane and solve the Lagrange–Euler equations, taking into account circularly polarized magnon states $l_\pm = l_x \pm i l_y$. The resulting coupled equations can be written as

$$\frac{d^2 l_+}{dt^2} + 2\zeta_m \frac{dl_+}{dt} + \left(\omega_0^2 - \gamma^2 H_{ext}^2\right) l_+ + 2\gamma i H_{ext} \frac{dl_+}{dt} = -2\alpha \theta_{ph} l_- + \gamma \frac{d}{dt}(h_y - i h_x),$$

(1)

$$\frac{d^2 l_-}{dt^2} + 2\zeta_m \frac{dl_-}{dt} + \left(\omega_0^2 - \gamma^2 H_{ext}^2\right) l_- - 2\gamma i H_{ext} \frac{dl_-}{dt} = 2\alpha \theta_{ph} l_+ + \gamma \frac{d}{dt}(h_y + i h_x),$$

(2)

$$\frac{d^2 \theta_{ph}}{dt^2} + 2\zeta_{ph} \frac{d\theta_{ph}}{dt} + \omega_{ph}^2 \theta_{ph} = -\alpha\left(l_+^2 + l_-^2\right),$$

(3)

where $\omega_i = 2\pi f_i$ and the Gilbert damping factors $\zeta_i$ with $i =$ "m" or "ph" account for the magnon or the phonon subsystem, correspondingly. The second term on the right-hand side of Eqs. (1)–(2) represents the linear Zeeman torque[34], while the nonlinear coupling can be introduced in Eqs. (1)–(3) as the derivative of $\Phi$ on the corresponding order parameter. These terms represent the mutual nonlinear perturbation of the magnon (phonon) subsystem by the phonon (magnon) subsystem. The experimental THz pulse waveform as measured by electro-optical sampling (see Fig. 2a) is introduced as the driving force in our simulation. The derivation of Eqs. (1)–(3) is provided in the Supplementary Material, Section B.

In Fig. 4a, b, simulated phonon spectra are plotted for external magnetic fields as applied in our experiment. Here, panel (a) corresponds to the weak coupling case with $\alpha = 0.07$, while panel (b) represents the case of the strong coupling with $\alpha = 7$. A drastic difference in the spectra for magnetic fields of $\mu_0 H_{ext} = 3$, 3.5, and 4 T can be seen, highlighting the role of the Fermi resonance. In the weak coupling regime of Fig. 4a, the spectra contain no splitting and the peak amplitude has only a single maximum close to $\mu_0 H_{ext} = 3.5$ T. The corresponding phonon weight plotted in panel (c) is also symmetric. In the strong coupling regime (see Fig. 4b), at $\mu_0 H_{ext} = 3.5$ T one observes a splitting of the phonon line resulting in a dip in the phonon weight. Moreover, the absolute peak of the phonon weight is shifted to 3 T. These features are well captured by our experimental data set (see Fig. 3) showing good agreement with our model. We elaborate on the explicit effect of our simulation parameters in the Supplementary Material, Section C.

## Discussion

Despite the fact that nonlinear phononics is a relatively young field, Fermi resonance of the lattice has been studied theoretically earlier[8–12] and can be seen as the nonlinear analog of the avoided crossing effect for strongly coupled modes[35,36]. In particular, it was shown that the resonances can result either in a splitting or broadening of phonon lines in the vibrational spectra of the lattice. Moreover, according to ref. 13, in the vicinity of the resonance, nonlinear damping plays a significant role and in principle can redistribute energy between the magnon and phonon subsystems.

In contrast to the previous theoretical studies focused on incoherent lattice dynamics and thus revealing stochastic acts of energy exchange between the modes[10,13], our experiments reveal the manifestation of the Fermi resonances for the case of coherent dynamics. The nature of mode interaction opens the possibility to control the modes' scattering rate, which, in conjunction with the pulsed excitation regime, provides a model system to study strong coupling interactions in the time domain. Thus, using the methods of coherent control one can, in general, steer the energy flow between spins and lattices on demand. We

propose that strong coupling results in a two magnon-phonon hybridization, paving the way to explore quantum effects of coupled dynamics in antiferromagnets. One of such intriguing aspect is the derivation of the magnon-phonon quasi-particles conservation law analogous to the Manley-Rowe relations for second harmonic generation[37].

In summary, we reported on a new regime of magnon-phonon dynamics, which in the vicinity of the Fermi resonance condition, facilitates a mutual, anharmonic energy exchange between magnons and phonons. We reveal that by tuning the eigenmode frequencies, we can control this process, enhancing magnon-phonon coupling. In particular, we demonstrate a broadening of phonon spectra and an asymmetric redistribution of the phonon weight upon tuning the magnon frequency with an external magnetic field. This suggests the formation of a strongly coupled two magnon-phonon hybridization state. We believe that our finding is among the first reports of non-linearly coupled magnon-phonon dynamics accompanied by a non-trivial energy exchange between the subsystems. Thus, our work represents an important milestone in the fields of magnonics and phononics[38], where coherent energy control plays a central role.

## Methods

### Material

In our experiment, we used a plane plate of $CoF_2$ single crystal where the tetragonal optical $c$-axis axis is oriented along the sample surface normally. The single crystal of $CoF_2$ was grown by the Bridgman method in platinum crucibles in a helium atmosphere as described in ref. [39]. The sample was cut from an X-ray oriented single crystal, prepared in the form of a plane parallel plate, and optically grade polished. The thickness of the plate was $d \approx 500\,\mu m$.

### Experimental technique

To achieve our goal of selectively pumping the magnon $f_m$, we have performed THz pump−IR probe spectroscopy of $CoF_2$ in magnetic fields up to $\mu_0 H_{ext} = 7$ T. To this end, we used a custom-developed wet split-coil 10 T magnet with optical access, commercially available from Oxford Instruments. The spectrally bright, accelerator based superradiant THz source (TELBE at the ELBE center for high-power radiation sources) centered at 1 THz with 20% bandwidth served as a pump while a synchronized table-top laser system (regenerative amplifier with a compressor) delivered NIR 40 fs probe pulses[31]. The oscillator feeding the amplifier was tightly synchronized to the accelerator via a home-built stabilized optical fiber link and a commercial Synchrolock-AP (Coherent Inc.) for fast phase correction. Slow "out-of-the-loop" timing drifts were continuously corrected by monitoring TELBE's single-cycle coherent diffraction radiator via spectral decoding and an additional mechanical delay stage. The peak electric field was estimated to be 100 kV/cm in free space. THz power was closely monitored and our data was normalized as described in the main text. To remove water absorption lines from the THz pump spectrum, we purged the THz beam path with nitrogen. A data acquisition card was used to record the signals. The polarization state of the probe pulse after the sample was monitored by the combination of a half-wave plate, Wollaston-prism, and carefully balanced photodetectors. The time domain trace of the THz pump pulse shown in Fig. 2a was obtained by electro-optical sampling[40] with a 2 mm thick ZnTe crystal.

### Data analysis

We performed the Fourier transformation on the full-time domain range of 130 ps shown in Fig. 2b and no additional filtering was applied to the data. As can be seen from ref. [24], the magnon scales linearly, and the phonon scales quadratic with THz electric field. Thus, we normalize the FFT range 0.5–1.5 THz (corresponding to magnon) by the square root of THz powers and the range 1.5–2.5 THz (corresponding to phonon) by THz powers, respectively. Subsequently, we extract the phonon weight in the narrow range from 1.9 to 2.0 THz (see Fig. 3b. This choice ensures that

no additional experimental noise is picked up in particular for $\mu_0 H_{ext} = 0$ T and 7 T where no phonon is present. We note that even for extracting the phonon peak amplitude, a clearly pronounced dip at $\mu_0 H_{ext} = 3.5$ T is present. However, we choose to extract the phonon spectral weight which ensures that not only the peak amplitude but also the line shape of the mode is taken into account.

### Simulation

Except otherwise specified, our simulation parameters are $\alpha = 7$, $\zeta_m/2\pi = 10$ GHz, and $\zeta_{ph}/2\pi = 0$. We performed our simulation over a time domain range equal to the full experimentally acquired range of 130 ps. The subsequent procedure for performing the Fourier transformation and extracting the phonon spectral weight by integration was performed identically to the experimental data treatment. To solve our differential equations, we apply a Runge–Kutta approximation of fourth order and employ the real THz pump pulse shape as shown in Fig. 2a. An extended discussion is provided in the Supplementary Material, Section C. From Fig. 2b, we extracted the phonon frequency $f_{ph} = 1.965$ THz, and the magnon frequency $f_m = f_0 - \gamma H_{ext}$ with gyromagnetic ratio $\gamma = 38.6$ GHz being the linear dependent on the external magnetic field $H_{ext}$.

### Theoretical considerations

See Supplementary Material, Section B for extended discussion.

### Reporting summary

Further information on research design is available in the Nature Portfolio Reporting Summary linked to this article.

## Data availability

All data are currently stored in the Springer Nature Figshare repository under https://doi.org/10.6084/m9.figshare.23578320.

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

## Acknowledgements

We thank Dr. Clément Faugeras for fruitful discussions. This work was supported by the Deutsche Forschungsgemeinschaft (DFG, German Research Foundation)—Project number 277146847—CRC 1238, de Nederlandse Organisatie voor Wetenschappelijk Onderzoek (NWO), the European Union's Horizon 2020 research and innovation program under the Marie Skłodowska-Curie grant agreement No. 861300 (COMRAD), the European Research Council ERC Grant Agreement No.101054664 273 (SPARTACUS), and RSF (grant No. 21-42-00035). MIK and AVK acknowledge the research program "Materials for the Quantum Age" (QuMat) for financial support. This program (registration number 024.005.006) is part of the Gravitation program financed by the Dutch Ministry of Education, Culture and Science (OCW). RMD acknowledges support of RSF (Grant No. 22-72-00025). Parts of this research were carried out at ELBE at the Helmholtz-Zentrum Dresden—Rossendorf e. V., a member of the Helmholtz Association. The TELBE measurements were conducted February 11–15, 2022 proposal number 21202639-ST, and October 23–25, 2021 proposal number 21102489-ST. The authors declare that this work has been published as a result of peer-to-peer scientific collaboration between researchers. The provided affiliations represent the actual addresses of the authors in agreement with their digital identifier (ORCID) and cannot be considered as a formal colla-boration between the aforementioned institutions.

## Author contributions

E.A.M. conceived the project; E.A.M., T.W.J.M., and A.V.K. coordinated the project; I.I., T.V.A.G.O., A.P., J.-C. D., S.K. set up the experiment at TELBE; R.V.P. and R.M.D. provided and characterized the samples; T.W.J.M., E.A.M., K.A.G., C.R., A.A., I.I., T.V.A.G.O., A.P., J.-C. D., S.K. performed the measurements at TELBE; T.W.J.M. analysed and visua-lized the results; A.V.K., P.H.M.L., M.I.K., and B.A.I. helped interpreting the results; B.A.I. provided the theory with input from M.I.K.; T.W.J.M. performed the numerical simulations; T.W.J.M. wrote the manuscript in close collaboration with E.A.M. and input from all co-authors.

## Competing interests

The authors declare competing interests.
