## [Peer Review File · Nature Communications]

Reviewers' Comments:

Reviewer #1:

Remarks to the Author:

The concept of quasiparticles has proven so successful that one is tempted to forget that it is based on weak interactions that can be treated as linear and harmonic. The authors remind us of this aspect by demonstrating nonlinearities in the interaction of the spins (\sim magnons) with the lattice (\sim phonons). They enhance the nonlinearity to the level of experimental detection by a doubly resonant two-photon excitation in the THz spectral range. In a combination of experiment and theory they demonstrate that the nonlinearity manifests itself as a double-peak feature in the magnon-driven phonon spectral weight.

I find this work simple and effective, almost didactical in nature, and therefore recommend publication in Nature Communications. It demonstrates the boundaries of an important physical picture and the value that nonlinearities have in many areas of science. I only have two suggestions that may improve the manuscript further, in particular for the non-specialist reader.

(1) The relation between nonlinearity and the double-peak feature in Fig. 3 is intriguing. I see the equations delivering this result, but I have trouble in finding an intuitive physical understanding of why there are two peaks. Can the authors give a qualitative explanation here?

(2) As the authors say, nonlinearities may be accessed via intense excitation or (this work) resonant excitation. Another route would be to make use of the "softness" of materials: small perturbations leading to exceptionally large responses. There have been recent works in this direction, and the authors might like to mention this.

Reviewer #2:

Remarks to the Author:

In the present manuscript Metzger et al. present an experiment where they explore the non-linear coupling between two distinct normal modes of an antiferromagnet. They use intense THz pulses supplied by a THz free-electron laser to resonantly excite a coherent magnon in the antiferromagnetic insulator CoF₂. They then follow the time-resolved polarization rotation of a weak near-infrared probe pulse to observe the coherent dynamics of the directly driven magnon and a non-linearly coupled phonon mode at about twice the magnon frequency f_M . Crucially, they use a static magnetic field to shift the magnon frequency to satisfy the so-called Fermi-resonance condition $2 f_M(H) = f_{ph}$. When this condition is met at a magnetic field of $H = 3$ T they find a resonant enhancement of the coherent phonon amplitude. They claim the phonon frequency f_{ph} remains unaffected by magnetic field. These experimental findings are corroborated by solving the coupled Lagrange-Euler equations which seem to be in good agreement with the magnetic field dependence of the phonon spectral weight.

The main findings of the paper can therefore be summarized as follows:

- 1) resonant enhancement of the coherent phonon spectral weight when $2 f_M(H) = f_{ph}$;
- 2) good agreement between their data and simulations based on non-linearly coupled Lagrange-Euler equations.

Based on this they claim to have observed a new regime of magnon-phonon dynamics and a new phenomenon in nonlinear phononics in general.

The data with its magnetic field dependence is of good enough quality and I believe the interpretation of the experimental data is reasonable. Yet, it is very difficult to grasp the overall message of the paper as the text is highly convoluted and not particularly easy to understand making it hard for a non-expert to follow. Further, certain very relevant references (such as arXiv:2207.07103) where similar/the same excitation mechanisms are being discussed are not included. Additionally, the word "impulsive" in the title is highly misleading as there is no impulsive

component to the observed dynamics (see below). Finally, the nonlinear mechanism described in this paper has already been investigated in an earlier work by a subset of the authors, therefore the claim of a “new phenomenon” made in this manuscript is not true. Therefore, I cannot recommend this paper for publication in Nature Communications and its broad readership. Overall, I cannot recommend the publication of this manuscript in the current form in any Journal. I will outline my reasons in the following.

Concerns regarding the data:

One of the main claims of the paper, the double resonance, hinges on one data point. To further solidify this claim, the authors should measure additional magnetic field values between 3 and 4 T. I also have serious misgivings about the generation of the data points in figure 3 a. It is not clear how these points were extracted. In the experimental portion of the manuscript the authors only write that they integrate “the area under the phonon spectra for different external magnetic fields Hext”. Later in the theoretical description they describe the use of two different time windows “where either the magnon or the phonon are dominating the dynamics” as was done in the experiment. Yet, this was only mentioned in this later part of the manuscript and the caption of figure 2. Please also add this to the main text. How would the results differ if not two but only one FFT was carried out over the whole time-window? From experience this can significantly alter the results and it needs to be checked. If only one FFT is performed Figure 2 b and c can be consolidated into one single panel.

Further, the text reads as if figure 2a shows the raw data measured in the experiment. This is indeed not the case (as is stated in the methods section). The authors need to make this extremely clear in the main text and the figure caption. It further seems like the data treatment introduced artifacts (oscillations) before time-zero as well as at later times. Was the data analysis (FFTs in figure 2b and c as well as the data in Figure 3a) obtained from the already filtered data. If yes, how do the results look like when performed on the raw data. This is extremely crucial considering that only one data point in Figure 3 a agrees with their claim of a “double resonance”. Did the authors measure an excitation field dependence. If they claim a non-linear coupling mechanism between the magnon and the phonon they should provide the reader with the necessary data to support this claim. This is of particular importance because to extract a meaningful amplitude of the phonon (for their magnetic field dependence in Figure 3a) they must excite the magnon to the same amplitude. In this light the statement in line 133 “which does not have to correlate directly with the magnon spectral amplitude” is extremely misleading and should be avoided. Figure 2b then shows that indeed the magnon frequency dramatically changes as a function of the magnetic field. For 0 T and 7 T the magnon amplitude is very small (because the excitation field has only negligible spectral components at the magnon frequency (see Fig A1 c)), and therefore the resulting phonon amplitude would also be expected to be smaller. This is independent of the Fermi condition. Therefore to obtain an unbiased magnetic field dependence the phonon amplitude would need to be normalized by the magnon amplitude squared. This problem can be circumvented by using either a tunable source or a broadband THz pulse. Why does the magnon suddenly disappear at a time-delay of 60 ps? (see figure A2) Why did the authors choose to conduct this experiment at a narrowband FEL facility rather than using a broadband tabletop source?

Concerns regarding the interpretation:

As mentioned above, the authors call their finding an “impulsive Fermi magnon-phonon resonance”. There is nothing impulsive about the dynamics observed here. For an impulsive excitation either the duration or the risetime of the stimulus must be faster than the period of the collective excitations. In the case of the present manuscript the excitation pulse has a duration of about 5 ps, this is too long to launch any impulsive dynamics. I would advise the authors to avoid calling their dynamics impulsive because all effects they are observing are due to either resonant pumping of the magnon or due to resonant ($2f_M = f_{ph}$) coupling of magnon and phonon. In line 127-128 the authors say that the data reveals the “forced response” of the magnon which “closely following the magnetic field of the THz pulse”. Comparing figure 2a and figure A1b shows that this statement is not true (pulse duration ~ 5 ps, duration of the dynamics ~ 20 -40 ps). Why would the excitation at 0 T or 7 T compared to 2-5T not lead to free, unforced oscillations of the Magnon?

The theoretical consideration using the Lagrange-Euler equations is sound. Yet, in the equations the authors did not include the direct electric field excitation of the Raman mode via E^2 . This term, like the magnon, should also be resonant because $2 f_{\text{THz}} = f_{\text{ph}}$. This is an oversight and should be accounted for.

The last paragraph of the theoretical description (line 182-190) is very confusing. They claim that only at larger values of the coupling term $\alpha > 3$ the reciprocal coupling from the phonon back to the magnon becomes seizable. This is not correct. This term is always in effect and even for smaller values ($\alpha = 2$) in their own simulation figure 3 b the shape of the phonon response is strongly distorted and does not resemble a Lorentzian (as expected for a uni-directional energy flow). The authors should simply avoid this discussion.

Albeit "Fermi resonance" might have more ring to it, the condition the authors are exploring here is more commonly known as a parametric resonance. I would appreciate if they could point it out in the paper or even adapt the title.

Minor Issues:

Since this is an experimental study I expect to see at least a sketch of the experimental geometry. I would advise the authors to move Figure A1 to the main text together with the crucial information of the temporal and spectral profile of the pump field.

The introductory paragraph is unnecessarily long. The authors should rather use the word count to write a cleaner demonstration of the experimental results.

Overall, the text is extremely hard to read. The authors should focus on using more concise language to make the text easier to read, even for an expert audience.

To conclude, I cannot recommend this paper for publication. Different to their claim, the presented physics are not new and are already published by the same authors in a different publication (DOI: 10.1126/science.abk1121). On top of that I have serious concerns about the analysis and data presentation as outline above.

REVIEWER COMMENTS

Reviewer #1 (Remarks to the Author):

The concept of quasiparticles has proven so successful that one is tempted to forget that it is based on weak interactions that can be treated as linear and harmonic. The authors remind us of this aspect by demonstrating nonlinearities in the interaction of the spins (~magnons) with the lattice (~phonons). They enhance the nonlinearity to the level of experimental detection by a doubly resonant two-photon excitation in the THz spectral range. In a combination of experiment and theory they demonstrate that the nonlinearity manifests itself as a double-peak feature in the magnon-driven phonon spectral weight.

I find this work simple and effective, almost didactical in nature, and therefore recommend publication in Nature Communications. It demonstrates the boundaries of an important physical picture and the value that nonlinearities have in many areas of science. I only have two suggestions that may improve the manuscript further, in particular for the non-specialist reader.

Reviewer (1.1): The relation between nonlinearity and the double-peak feature in Fig. 3 is intriguing. I see the equations delivering this result, but I have trouble in finding an intuitive physical understanding of why there are two peaks. Can the authors give a qualitative explanation here?

Response (1.1): We thank the Reviewer for the encouraging remarks and the constructive feedback.

In response to the criticism, we have provided a qualitative explanation of the origin of the double-peak feature. In fact, the double peak line shape can be seen as a result of the nonlinear analogue to the “avoided crossing effect” between a two-magnon state and a phonon state (see, for instance, Fig.4 in J. Appl. Phys. 134, 201101 (2023)). Since the phonon and the two-magnon state have the very same symmetry, and thus can be hybridized, their excitation cannot be considered separately. Our experimental results supported by simulations show that as a result of the avoided crossing at the magnon-phonon Fermi resonance, the phonon spectral line (a) reduces in amplitude, (b) broadens and (c) eventually splits at $H_{\text{ext}} = 3.5\text{T}$. Moreover, the extracted phonon spectral weight as function of the applied magnetic field shows an overall asymmetric line shape. This is in excellent agreement with our straightforward model predicting line-shapes strikingly similar to those observed experimentally. Following the suggestions of Reviewer 1, we have expanded our main text as well as the conclusion and discussion sections for clarity of physical interpretation, see Summary of changes: **Issue 1**

Reviewer (1.2): As the authors say, nonlinearities may be accessed via intense excitation or (this work) resonant excitation. Another route would be to make use of the "softness" of materials: small perturbations leading to exceptionally large responses. There have been recent works in this direction, and the authors might like to mention this.

Response (1.2): We fully agree with the Reviewer. In response to the criticism, we have cited papers reporting about alternative routes to nonlinearities, including the route employing the “softness” of materials. For instance, [Ozhogin et al. Anharmonicity of mixed modes and giant acoustic nonlinearity of antiferromagnetics *Sov. Phys. Usp.* 31 713–729 (1988)].

Reviewer #2 (Remarks to the Author):

In the present manuscript Metzger et al. present an experiment where they explore the non-linear coupling between two distinct normal modes of an antiferromagnet. They use intense

THz pulses supplied by a THz free-electron laser to resonantly excite a coherent magnon in the antiferromagnetic insulator CoF₂. They then follow the time-resolved polarization rotation of a weak near-infrared probe pulse to observe the coherent dynamics of the directly driven magnon and a non-linearly coupled phonon mode at about twice the magnon frequency f_M . Crucially, they use a static magnetic field to shift the magnon frequency to satisfy the so-called Fermi-resonance condition $2 f_M(H) = f_{ph}$. When this condition is met at a magnetic field of $H = 3$ T they find a resonant enhancement of the coherent phonon amplitude. They claim the phonon frequency f_{ph} remains unaffected by magnetic field. These experimental findings are corroborated by solving the coupled Lagrange-Euler equations which seem to be in good agreement with the magnetic field dependence of the phonon spectral weight.

The main findings of the paper can therefore be summarized as follows:

- 1) resonant enhancement of the coherent phonon spectral weight when $2 f_M(H) = f_{ph}$;
- 2) good agreement between their data and simulations based on non-linearly coupled Lagrange-Euler equations.

Based on this they claim to have observed a new regime of magnon-phonon dynamics and a new phenomenon in nonlinear phononics in general. The data with its magnetic field dependence is of good enough quality and I believe the interpretation of the experimental data is reasonable. Yet, it is very difficult to grasp the overall message of the paper as the text is highly convoluted and not particularly easy to understand making it hard for a non-expert to follow.

Reviewer (2.1): Further, certain very relevant references (such as arXiv:2207.07103) where similar/the same excitation mechanisms are being discussed are not included.

Response (2.1): We thank the Reviewer for letting us know about a very relevant upcoming paper. We have cited the paper together with a more recent work of the same authors [arXiv:2301.12555]. Although this is a very relevant reference, we respectfully disagree that the discussed mechanism is similar to or the same as in our paper. For instance, magnetic field dependence, being the core of our work, is not discussed in [arXiv:2207.07103].

Reviewer (2.2): Additionally, the word “impulsive” in the title is highly misleading as there is no impulsive component to the observed dynamics (see below).

Response (2.2): By ‘impulsive’, we mistakenly tried to describe the pulsed origin of the observed phenomena and agree that it might mislead interpretation. We are sorry about this and updated the title accordingly. The updated title reads “Magnon-phonon Fermi resonance in antiferromagnetic CoF₂”. We emphasize the importance of a pulsed excitation compared to cw-excitation in the main text.

Reviewer (2.3): Finally, the nonlinear mechanism described in this paper has already been investigated in an earlier work by a subset of the authors, therefore the claim of a “new phenomenon” made in this manuscript is not true.

Response (2.3): We strongly disagree. Although the present work was inspired by our earlier paper [Science 374, 1608-1611 (2021)], the conditions of the magnon-phonon Fermi resonance have been realized and studied for the first time in the present work. To achieve these conditions, we needed a unique experimental facility facilitating narrow-band resonant pumping of the 1 THz magnon mode and magnetic fields up to 7 T. Only with fine tuning of both the light frequency and magnetic field available, it became possible to demonstrate the peculiar features in the magnetic field dependence which can be explained intuitively as the result of an efficient bi-directional magnon-phonon energy exchange. The latter can also be

interpreted as a nonlinear analogue to the “avoided crossing effect” between the two-magnon and the phonon states, see Response (1.1) to Reviewer 1. Moreover, the mechanism of the observed phonon amplitude dependence on external magnetic fields is distinctly different from those reported so far [Nano Lett. 20, 5991–5996 (2020), Phys. Rev. Lett. 128, 075901 (2022), J. Phys. C 9, L297 (1976)].

Reviewer (2.4): Therefore, I cannot recommend this paper for publication in Nature Communications and its broad readership.

Response (2.4): We believe that the conclusion of the Reviewer 2 is a result of misunderstanding. We hope that the Reviewer 2 will find our arguments convincing and we thank the Reviewer 2 for the stimulating criticism.

Reviewer (2.5): Overall, I cannot recommend the publication of this manuscript in the current form in any Journal. I will outline my reasons in the following. Concerns regarding the data: One of the main claims of the paper, the double resonance, hinges on one data point. To further solidify this claim, the authors should measure additional magnetic field values between 3 and 4 T.

Response (2.5): We are afraid that the Reviewer has overlooked that the magnon-phonon Fermi resonance realized and studied in our work manifests itself in by far more than a single data point. Our experimental results supported by simulations show that at the magnon-phonon Fermi resonance the phonon spectral line (a) reduces in amplitude, (b) broadens and (c) eventually splits at $H_{\text{ext}} = 3.5\text{T}$. Moreover, the extracted phonon spectral weight as function of the applied magnetic field shows an overall asymmetric line shape. In the current version of our manuscript, we extend the discussion on these observed peculiarities, see Summary of changes: **Issue 1**.

We would like to express our disappointment that Reviewer 2 has overlooked that obtaining the reported phenomena could only become possible by fine tuning of the frequency of a narrow-band THz excitation and high external magnetic fields. Such a combination is absolutely unique and so far available only at the large-scale free-electron laser facility HZDR. Furthermore, the high-repetition operation rate of a few hundred kilohertz, coupled with a high-quality data acquisition technique, demonstrates exceptional sensitivity in detection. We thus find the requests of the Reviewer to perform additional measurements not only unnecessary, but even impossible to realize on any timescale shorter than a year.

Reviewer (2.6): I also have serious misgivings about the generation of the data points in figure 3 a. It is not clear how these points were extracted. In the experimental portion of the manuscript the authors only write that they integrate “the area under the phonon spectra for different external magnetic fields H_{ext} ”.

Response (2.6): We would like to express our concerns that here, the Reviewer does not quote the full statement leaving out essential parts. In fact, the previous version of our manuscript, in particular the caption of Fig. 2 reads “The shaded area corresponds to the integration region of interest for extraction of the spectral weight”. In Fig. 2c, the area under the phonon line was shaded in colour thus clearly defining the integration area. Hence, we must, unfortunately, state that the criticism is not adequate here.

Reviewer (2.7): Later in the theoretical description they describe the use of two different time windows “where either the magnon or the phonon are dominating the dynamics” as was done in the experiment. Yet, this was only mentioned in this later part of the manuscript and the caption of figure 2. Please also add this to the main text. How would the results differ if not

two but only one FFT was carried out over the whole time-window? From experience this can significantly alter the results and it needs to be checked. If only one FFT is performed Figure 2 b and c can be consolidated into one single panel.

Response (2.7): In the current version of our manuscript, we take the Fourier transformation of the full time domain data (130ps). Importantly, magnon-phonon Fermi resonance as manifested by overall asymmetry and at dip at $H_{\text{ext}} = 3.5\text{T}$ of the extracted phonon weight vs external magnetic field, remains present and independent of filtering, Fourier transformation windows and other technicalities. As suggested by the Reviewer, we consolidated the FFTs into one single panel, see Fig. 3a. We have included an extended discussion in the methods and the Supplementary section; please refer to the Summary of Changes: **Issue 1, 2 and 7**

Reviewer (2.8): Further, the text reads as if figure 2a shows the raw data measured in the experiment. This is indeed not the case (as is stated in the methods section). The authors need to make this extremely clear in the main text and the figure caption. It further seems like the data treatment introduced artifacts (oscillations) before time-zero as well as at later times. Was the data analysis (FFTs in figure 2b and c as well as the data in Figure 3a) obtained from the already filtered data. If yes, how do the results look like when performed on the raw data. This is extremely crucial considering that only one data point in Figure 3 a agrees with their claim of a “double resonance”.

Response 2.8: In response to the criticism, we have changed Fig. 2. Now it shows the raw data without low-pass filtering. Importantly, magnon-phonon Fermi resonance as manifested by overall asymmetry and at dip at $H_{\text{ext}} = 3.5\text{ T}$ of the extracted phonon weight vs external magnetic field, remains present and independent of filtering, Fourier transformation windows and other technicalities. We emphasize that all data treatment in the main text of our current manuscript has been done without pre-filtering the time domain data. We believe the ‘artefacts’ mentioned by Reviewer 2 refer to the peculiarity of the superradiant THz pump source. The full time trace of our THz pump pulse can be found in Fig. 2(a) of the main text.

We note that data treatment as in the first version of our manuscript leads to even more pronounced features of magnon-phonon Fermi resonance. In particular, pre-filtering of the time domain data by a low-pass filter with cut-off frequency at 2.5 THz removes high-frequency noise introduced by the data acquisition scheme. To illustrate this, we provided Supplementary Fig. A1(c) of which a duplicate is shown below. Here, we demonstrate an excellent agreement of time domain data and sinusoidal fits to the magnon (red) and phonon (blue) frequencies. In particular, we highlight that in the region of 70-120ps exclusively an undamped phonon mode is present.

Figure 1: Duplicate of Supplementary Fig. A1(c). Low-pass filter with cut-off at 2.5THz applied to the raw data (grey solid line). The red (blue) solid line corresponds to a sinusoidal fit at the magnon (phonon) frequency.

Reviewer (2.9): Did the authors measure an excitation field dependence. If they claim a non-linear coupling mechanism between the magnon and the phonon they should provide the reader with the necessary data to support this claim. This is of particular importance because to extract a meaningful amplitude of the phonon (for their magnetic field dependence in Figure 3a) they must excite the magnon to the same amplitude. In this light the statement in line 133 “which does not have to correlate directly with the magnon spectral amplitude” is extremely misleading and should be avoided.

Response (2.9): We stress that, in accordance with the law of conservation of energy, any possible linear mechanism cannot explain that the phonon at the frequency of 1.96 THz is excited by the narrow-band 1 THz pump pulse employed in our experiment. Thus, the phonon can only be excited via a nonlinear mechanism. To avoid confusion, we have adapted the aforementioned statement, please see Summary of changes: **Issue 3**.

Reviewer (2.10): Figure 2b then shows that indeed the magnon frequency dramatically changes as a function of the magnetic field. For 0 T and 7 T the magnon amplitude is very small (because the excitation field has only negligible spectral components at the magnon frequency (see Fig A1 c)), and therefore the resulting phonon amplitude would also be expected to be smaller. This is independent of the Fermi condition. Therefore to obtain an unbiased magnetic field dependence the phonon amplitude would need to be normalized by the magnon amplitude squared.

Response (2.10): We agree with the Reviewer that for $H_{\text{ext}}=0\text{T}$ and $H_{\text{ext}}=7\text{T}$, the magnon amplitude is small due to negligible spectral components of the THz pump pulse at the respective magnon frequencies. However, we can only partly agree with Reviewer 2 on the suggested normalization procedure.

Firstly, normalizing at $H_{\text{ext}}=0\text{ T}$ and $H_{\text{ext}}=7\text{ T}$ fields is challenging, as no phonon response is visible at this frequency. Meanwhile, in the most interesting range of 2-5 T, magnon amplitude remains partially constant, and the dependence on Ph/M^2 would resemble the one plotted in the main text. However, we believe that was crucial to monitor the THz power dependence and this is exactly what we did. Thus, we normalize the data shown in Fig. 3 from 0.5-1.5 THz (magnon) by the square root of THz power and the data shown from 1.8-

2.1 THz (phonon) by the THz power. We believe that this way of normalizing our data provides a better insight and hope that the Reviewer will share our opinion.

Reviewer (2.11): This problem can be circumvented by using either a tunable source or a broadband THz pulse. Why does the magnon suddenly disappear at a time-delay of 60 ps? (see figure A2) Why did the authors choose to conduct this experiment at a narrowband FEL facility rather than using a broadband tabletop source?

Response (2.11): We respectfully disagree with the Reviewer. As mentioned above on several occasions, we are convinced that the reported physics could only be revealed using the unique specifications of the facility at HZDR. In particular, to demonstrate the nonlinear mechanism of 2 THz phonon excitation, we needed a fine-tunable narrow-band source of intense THz radiation that pumps exclusively the 1 THz magnon line. Simply speaking, this completely rules out any speculation regarding the direct (i.e. linear) excitation of the phonon by the THz pump pulse. The magnon disappears as a result of decoherence following the law of exponential decay. We do not see any sudden changes in the magnon amplitude at 60 ps, but can admit, however, that upon a decrease of the amplitude below the noise level the disappearance may look sudden.

Concerns regarding the interpretation:

Reviewer (2.12): As mentioned above, the authors call their finding an “impulsive Fermi magnon-phonon resonance”. There is nothing impulsive about the dynamics observed here. For an impulsive excitation either the duration or the rise time of the stimulus must be faster than the period of the collective excitations. In the case of the present manuscript the excitation pulse has a duration of about 5 ps, this is too long to launch any impulsive dynamics. I would advise the authors to avoid calling their dynamics impulsive because all effects they are observing are due to either resonant pumping of the magnon or due to resonant ($2f_M = f_{ph}$) coupling of magnon and phonon.

Response (2.12): We have responded to the criticism of Reviewer 2 above, see Response (2.2).

Reviewer (2.13): In line 127-128 the authors say that the data reveals the “forced response” of the magnon which “closely following the magnetic field of the THz pulse”. Comparing figure 2a and figure A1b shows that this statement is not true (pulse duration ~5ps, duration of the dynamics ~20-40 ps). Why would the excitation at 0 T or 7 T compared to 2-5T not lead to free, unforced oscillations of the Magnon?

Response (2.13): We admit that next to the dominating forced response, one cannot ignore much weaker dynamics observed in the 20-40 ps range. Note that the magnon dynamics are excited very inefficiently since the magnon spectra at $H_{ext}=0$ T or $H_{ext}=7$ T only weakly overlap with the THz pump spectrum, the signal in the window of 20-40 ps is additionally affected by multiple reflections of the pump pulse inside the crystal. To avoid further misunderstandings, we have adapted the corresponding lines in the main text. Please refer to the Summary of Changes: **Issue 4**.

Reviewer (2.14): The theoretical consideration using the Lagrange-Euler equations is sound. Yet, in the equations the authors did not include the direct electric field excitation of the Raman mode via E^2 . This term, like the magnon, should also be resonant because $2 f_{THz} = f_{ph}$. This is an oversight and should be accounted for.

Response (2.14): We would like to stress that no indication of the presence of an E^2 Raman term has been observed in the experiment. In particular, far away from the Fermi resonance condition at 0T and 7T where a forced magnon response dominates the outcome of the

experiment, we cannot see a contribution from the E^2 term. To be concise, we followed the advice of the Reviewer 2 and included this term in our simulations. As can be seen from the graph shown below, this term remains unaffected by external magnetic fields. However, we acknowledge that the offset for data points of $H_{\text{ext}}=0$ and $H_{\text{ext}}=7\text{T}$ in Fig. 3(b) might be interpreted as correlation with the E^2 term.

Figure 2: (a) Simulated phonon spectral weight vs external magnetic field for E^2 and L^2 terms and (b) Simulated phonon spectral weight with L^2 term vs experimentally extracted value.

Reviewer (2.15): The last paragraph of the theoretical description (line 182-190) is very confusing. They claim that only at larger values of the coupling term $\alpha > 3$ the reciprocal coupling from the phonon back to the magnon becomes seizable. This is not correct. This term is always in effect and even for smaller values ($\alpha = 2$) in their own simulation figure 3 b the shape of the phonon response is strongly distorted and does not resemble a Lorentzian (as expected for a uni-directional energy flow). The authors should simply avoid this discussion.

Response (2.15): In response to the criticism, we have adapted our discussion on the importance of coupling strength between the magnon and phonon subsystems. Specifically, we plot a pair of phonon Fourier spectra corresponding to weak and strong coupling cases in the main text. These spectra, along with the extracted phonon spectral weights, are presented in the current Fig. 4. Here, the fingerprints of magnon-phonon Fermi resonance are manifested by a broadening of the phonon line shape at $H_{\text{ext}}=3.5\text{T}$ and a dip in the phonon spectral weight which arise only for strong coupling with a nonlinear coupling term of $\alpha=7$. Please, see Summary of changes: **Issue 5**.

Reviewer (2.16): Albeit “Fermi resonance” might have more ring to it, the condition the authors are exploring here is more commonly known as a parametric resonance. I would appreciate if they could point it out in the paper or even adapt the title.

Response (2.16): We respectfully disagree with Reviewer. The more commonly known phenomena of parametric resonance describe two coupled and *directly* excited subsystems. The mathematical structure of coupled Eqs. 1-3 is distinctly different from the equations governing parametric resonance. However, our observation undoubtedly deals with a direct excitation of only the magnon resonance by a narrow band high intense THz pulse. We are convinced that

here we do apply a conventionally accepted terminology (see, for instance, J. Appl. Phys. 134, 201101 (2023)).

Minor Issues:

Reviewer (2.17): Since this is an experimental study I expect to see at least a sketch of the experimental geometry. I would advise the authors to move Figure A1 to the main text together with the crucial information of the temporal and spectral profile of the pump field.

Response (2.17): Following the suggestion of the Reviewer, we have moved a sketch of the experimental geometry along with important information about the temporal and spectral profile of the pump pulse from the supplementary material to the main text.

Reviewer (2.18): The introductory paragraph is unnecessarily long. The authors should rather use the word count to write a cleaner demonstration of the experimental results. Overall, the text is extremely hard to read. The authors should focus on using more concise language to make the text easier to read, even for an expert audience.

Response (2.18): We respect the opinion of Reviewer 2, but must note that Reviewer 1 does find our work “simple and effective, almost didactical in nature”. In response to the criticism, we have adapted and strengthened our arguments. We have also introduced substantial changes in the introductory paragraph, see Summary of changes: *Issue 6*.

Reviewer (2.19): To conclude, I cannot recommend this paper for publication. Different to their claim, the presented physics are not new and are already published by the same authors in a different publication (DOI: 10.1126/science.abk1121). On top of that I have serious concerns about the analysis and data presentation as outline above.

Response (2.19): We strongly disagree with Reviewer 2 and emphasize that the magnon-phonon Fermi resonance phenomena presented in this article is novel, has not been studied before and could only be studied at the unique large-scale facility HZDR. We refer to the updated version of our manuscript, the aforementioned explanations and hope that all Reviewers will share our opinion.

Summary of changes

Issue 1

Main text / Results

p. 4-5, lines 120-148

Performing Fourier transformation of the whole time domain range (-10 to 120 ps) in Fig. 3(a) reveals the presence of the magnon response oscillating at frequency f_m , and a second, distant peak at the B_{1g} phonon frequency f_{ph} . For magnetic fields of $\mu_0 H_{ext} = 0$ T and $\mu_0 H_{ext} = 7$ T the THz pump spectrum barely covers the magnon mode and THz induced polarization rotation contains a substantial contribution of the spectrally broad forced magnetic response closely following the magnetic field of the THz pulse, see Fig. 2 (b). Moreover, no phonon-induced dynamics are observed at these fields, implying that nonlinear excitation of phonons via the mechanism described in Ref. [27] does not play a significant role here. Closer to the Fermi resonance for the in-between magnetic fields of $\mu_0 H_{ext} = 2 - 5$ T, we observe the low energy magnon branch f_m with its frequency linearly decreasing with external magnetic field. Remarkably, the strongest magnon peak at $\mu_0 H_{ext} = 5$ T does not correspond to the strongest phonon peak, revealing complex dynamics in the vicinity of the magnon-phonon Fermi resonance. The most peculiar feature is observed at $\mu_0 H_{ext} = 3.5$ T, see Fig. 3(a). Firstly, the phonon peak amplitude for the dynamics at this magnetic field

is substantially reduced with respect to the peak amplitudes for $\mu_0 H_{ext} = 3$ T and $= 4$ T. Secondly, the phonon spectrum at $\mu_0 H_{ext} = 3.5$ T becomes broader. In fact, this resembles a splitting of the spectral line reported for purely phononic [10] or purely magnonic [15] systems under continuous wave pumping in vicinity of their Fermi resonances. To capture the energy redistribution, we integrate the area under the phonon spectra for different external magnetic fields H_{ext} over the shaded range of 1.9 THz to 2.0 THz, and retrieve the behaviour of the phonon resonance curve as shown in Fig. 3(b). Here, the phonon resonance line is clearly asymmetric with a pronounced dip at $\mu_0 H_{ext} = 3.5$ T indicating non-trivial magnon-phonon energy exchange. In the following section, we assign this feature to the unique benchmarks of magnon-phonon Fermi resonance.

Issue 2

Supplementary / Appendix A Additional experimental data (full section)

and

Main text / Methods / Data analysis (p. 9-10, lines 261-273)

We performed the Fourier transformation on the full time domain range of 130 ps shown in Fig. 2(b) and no additional filtering was applied to the data. As can be seen from [18], the magnon scales linearly and the phonon scales quadratic with THz electric field. Thus, we normalize the FFT range 0.5 - 1.5 THz (corresponding to magnon) by square-root of THz powers and the range 1.5 - 2.5 THz (corresponding to phonon) by THz powers, respectively. Subsequently, we extract the phonon weight in the narrow range from 1.9 to 2.0 THz (see Fig. 3(b). This choice ensures that no additional experimental noise is picked up in particular for $\mu_0 H_{ext} = 0$ T and 7 T where no phonon is present. We note that even for extracting the phonon peak amplitude, a clearly pronounced dip at $\mu_0 H_{ext} = 3.5$ T is present. However, we choose to extract the phonon spectral weight which ensures that not only the peak amplitude but also the line shape of the mode is taken into account.

Issue 3

Main text / Results (p. 4-5, lines 132-134)

Remarkably, the strongest magnon peak at $\mu_0 H_{ext} = 5$ T does not correspond to the strongest phonon peak, revealing complex dynamics in the vicinity of the magnon-phonon Fermi resonance.

Issue 4

Main text / Results (p. 4, lines 123-127)

For magnetic fields of $\mu_0 H_{ext} = 0$ T and $\mu_0 H_{ext} = 7$ T the THz pump spectrum barely covers the magnon mode and THz induced polarization rotation contains a substantial contribution of the spectrally broad forced magnetic response closely following the magnetic field of the THz pulse, see Fig. 2 (b).

Issue 5

Main text / Results (p. 8, lines 181-194)

In Figure 4(a-b), simulated phonon spectra are plotted for external magnetic fields as applied in our experiment. Here, panel (a) correspond to the weak coupling case with $\alpha = 0.07$, while panel (b) represents the case of the strong coupling with $\alpha = 7$. A drastic difference in the spectra for magnetic fields of $\mu_0 H_{ext} = 3, 3.5$ and 4 T can be seen, highlighting the role of the Fermi resonance. In the weak coupling regime of Fig. 4(a), the spectra contain no splitting and the peak amplitude has only a single maximum close to $\mu_0 H_{ext} = 3.5$ T. The corresponding phonon weight plotted in panel (c) is also symmetric. In the strong coupling regime (see Fig.

4(b)), at $\mu_0 H_{ext} = 3.5 T$ one observes a splitting of the phonon line resulting in a dip in the phonon weight. Moreover, the absolute peak of the phonon weight is shifted to 3 T. These features are well captured by our experimental data set (see Fig. 3) showing good agreement with our model. We elaborate on the explicit effect of our simulation parameters in the Supplementary material, section C.

Issue 6

Main text / Introduction (p. 2-3, lines 59-78)

In general, such behavior is expected between any eigenmodes x and y that involve a nonlinear coupling term x^2y [17]. Interestingly, the recently demonstrated nonlinear excitation of a phonon mode mediated by a magnon state suggests the presence of such a term in antiferromagnetic CoF_2 [18]. We anticipate that in the vicinity of the resonance, the nonlinear magnon-phonon interaction will affect the coupled dynamics dramatically. However, in zero applied magnetic field the double magnon frequency $2f_0$ of at $T = 6 K$ is higher than the frequency f_{ph} of the B_{1g} phonon $2f_0 > f_{ph}$ [19] and thus the system is not in resonance (see Fig. 1(a)). This is why we propose to apply an external magnetic field along the magnetic easy axis, which splits the magnon branches, while the phonon frequency remains unchanged [20]. Particularly, a field of $\mu_0 H_{ext} = 4 T$ is expected to be sufficient to reach the frequency matching condition (Fig. 1(a), grey star) with a lower energy magnon branch $2f_m = f_{ph}$, where the conditions of Fermi resonance might be satisfied. In Fig. 1(b), we illustrate the aforementioned processes by a graphical representation for an off-resonant system (I) and the magnon-phonon subsystem under Fermi-resonance condition (II). A pictorial representation of the energy transfer is depicted by Feynman diagrams in Fig. 1(c), illustrating the magnon-phonon interaction.

Main text / Results (see Issue 1, 2)

Main text / Discussion (full section)

Main text / Conclusion (full section)

Reviewers' Comments:

Reviewer #1:

Remarks to the Author:

The authors fully clarified the two remaining questions I had. My only final recommendation is to delete the lines connecting the data points in Figure 3b because they are misleading to the eye and strictly speaking incorrect because there is no linear connection between the points.

I also found the discussion with the other reviewer instructive. While, to my impression, the authors defended their claim of suitability for Nature Communications in a convincing way, the reviewer uncovered several unclaritys in the report. Again, to my view, the authors responded well to these and significantly improved their manuscript.

Reviewer #2:

Remarks to the Author:

Metzger et al. present a revised manuscript, where they addressed some of the issues I raised in the initial review of their work. Unfortunately, I don't not find their revisions and arguments convincing enough to recommend publication of this manuscript in Nature Communications. The main reason of my initial rejection of this manuscript in a journal with a diverse readership such as Nature Communications is that the finding is not novel and rather addresses a different aspect of the same observation in [Science 374, 1608-1611 (2021)].

The authors argue that the conditions of the 'magnon-phonon Fermi resonance have been realized and studied for the first time in the present work' thanks due to the unique capabilities of the HZDR and its THz-FEL combined with magnetic field up to 7 T. Yet, not only were the authors able to observe the magnon-phonon coupling in their previous work [Science 374, 1608-1611 (2021)] but they indeed observed the 'magnon-phonon Fermi resonance' in this work as well (see Figure 3B off that paper). In this paper, however, they did not need to use the unique capabilities of the HZDR but rather carried the experiments out with a broadband tabletop THz source and instead of tuning the magnon frequency with a magnetic field they did so using temperature. Here as well they observed a resonant enhancement of the phonon amplitude when the condition $2 f_M(T) = f_{ph}$ is fulfilled. This is why I believe the author's claim that the HZDR facility is tantamount for this experiment and that the physics are novel is not true. I do agree with the authors that the THz facility at the HZDR is an incredibly valuable tool, yet I also think it is the wrong tool for the presented scientific case and no new insight is gained.

Further, I still believe that the conclusion of a splitting phonon resonance hinges on the one data point at 3.5 T magnetic field. It is hard to judge that the splitting is real or just due to poor signal to noise (error bars are missing as well to assess this better). They further claim that "the magnon-phonon Fermi resonance the phonon spectral line (a) reduces in amplitude, (b) broadens and (c) eventually splits at $H_{ext} = 3.5T$." I find this hard to see from figure 3a and can only stress again that the FFT line shape strongly depends on the FFT time windows. Also, I want to state here that this was my main concern about how the data points in Figure 3b were obtained (in the rebuttal Reviewer (2.6)), which the authors responded to in (Reviewer (2.7)).

Another, major shortcoming of this work is the varying excitation density for different magnon frequencies (different magnetic fields). As observed by the authors this dramatically influences the observed amplitude of the magnon (Fig 3a). To address this, they "normalized" their data by the incident THz power (phonon) and the square root of THz power (magnon) (The THz power is nowhere to be found in the manuscript however). Unfortunately, this data treatment is not correct. For every data set (as function of magnetic field), they have to normalize the (phonon) Magnon amplitude by the (square of the) FFT amplitude at the magnon frequency or control the incident THz power in order to ensure equal magnon excitation densities. This has not been done. As I

pointed out in my last review: "This problem can be circumvented by using either a tunable source or a broadband THz pulse." Sources capable to conduct this experiment in the "home" lab exist with respect of repetition, rate, frequency tunability and intensity. Therefore, I find their arguments not credible.

Beyond these major issues there are other aspects where I disagree with the statements of the authors. Unfortunately, since my last review very little has changed, and I am not convinced that this manuscript should be published in Nature Communications. Yet, I believe with more revisions this paper could be suitable for a more specialized journal.

Reviewer #3:

Remarks to the Author:

The authors present a study of nonlinear magnon-phonon dynamics in antiferromagnetic CoF₂ for the case where the frequency of the magnon is nearly half the frequency of the phonon. By applying a magnetic field, the authors tune the frequency of the magnon into half-frequency resonance with the phonon mode, thereby strongly enhancing the energy transfer of the sum-frequency excitation process. The authors interpret this as a manifestation of Fermi resonance, supported by their measurements of change in spectral shape and intensity of the phonon.

In my opinion, this is an exciting study demonstrating a new mechanism of nonlinear magnon-phonon excitation, which deserves publication in Nature Communications.

The editor asked me to comment on the response by the authors to the reviews and in particular on point 2.5 of the response. In my opinion, the data is sufficient to support the claim of a magnon-phonon Fermi resonance. A direct 2-photon driving force (E^2) can be excluded due to the magnetic-field dependent spectral weight of the photon (Fig. 2 in the response). This further excludes a 1-photon-1-magnon driving force ($E \cdot I$) as previously demonstrated by some of the authors, and leaves only the 2-magnon driving force (I^2) as a possible source of the phonon excitation. Overall, I believe that the authors responded comprehensively to the comments by the reviewers and made appropriate changes to their manuscript.

Further questions and comments:

- The 2-magnon-1-phonon scattering processes shown in Fig. 1c of the manuscript correspond to magnonic sum-frequency excitation of the phonon (left) and parametric downconversion of the phonon to two magnons (right), which the authors demonstrate here for the first time. What distinguishes the Fermi resonance the authors see from a simple combination of these two mechanisms? Figs. 4a,b suggests that the splitting of the phonon spectral weight occurs only at larger coupling strengths. Can the Fermi resonance therefore be seen as the strong-coupling regime of sum-frequency excitation and parametric downconversion?

- The authors point out in the discussion that nonlinear phononics is still a young field. A lot of work has been done in recent years for scattering processes related to that presented by the authors here however, and I think it would be worth connecting to some of them. For example, sum-frequency excitation in a 2-phonon-1-magnon scattering process has been reported recently [PRB 103, 094407 (2021)]. Furthermore, sum-frequency excitations for 2-phonon-1-phonon scattering processes have become increasingly popular in the field, including a recent demonstration by some of the authors, e.g. [PRL 131, 026902 (2023)], [PRB 97, 174302 (2018)], [PRB 102, 224301 (2020)], and [PRB, 97, 214304 (2018)]. In addition, the concept of enhancing a sum-frequency excitation process by tuning a quasiparticle frequency into a half-frequency resonance (here achieved by applying a magnetic field for the magnon) has been explored for a 2-phonon-1-phonon mechanism previously using optical cavities [PRResearch 3, L032046 (2021)].

Since the authors state in the introduction that the Fermi resonance is expected for any type of x^2y coupling, this would make their findings generalizable to the strong-coupling regimes of the above coupling mechanisms as well.

Minor correction: Refs. [1] and [27] in the manuscript are the same.

To conclude, I recommend publication after addressing the above points.

REVIEWER COMMENTS

Reviewer #1 (Remarks to the Author):

Reviewer (1.1): The authors fully clarified the two remaining questions I had. My only final recommendation is to delete the lines connecting the data points in Figure 3b because they are misleading to the eye and strictly speaking incorrect because there is no linear connection between the points. I also found the discussion with the other reviewer instructive. While, to my impression, the authors defended their claim of suitability for Nature Communications in a convincing way, the reviewer uncovered several unclarities in the report. Again, to my view, the authors responded well to these and significantly improved their manuscript.

Response (1.1): We would like to express our appreciation for the positive validation of our manuscript by Reviewer 1 and for the comments that have further helped us to improve our work. Following the recommendation of the Reviewer, we have updated Fig. 3b in the revised version of the manuscript.

Reviewer #2 (Remarks to the Author):

Reviewer (2.1): Metzger et al. present a revised manuscript, where they addressed some of the issues I raised in the initial review of their work. Unfortunately, I don't find their revisions and arguments convincing enough to recommend publication of this manuscript in Nature Communications. The main reason of my initial rejection of this manuscript in a journal with a diverse readership such as Nature Communications is that the finding is not novel and rather addresses a different aspect of the same observation in [Science 374, 1608-1611 (2021)]. The authors argue that the conditions of the 'magnon-phonon Fermi resonance have been realized and studied for the first time in the present work' thanks due to the unique capabilities of the HZDR and its THz-FEL combined with magnetic field up to 7 T. Yet, not only were the authors able to observe the magnon-phonon coupling in their previous work [Science 374, 1608-1611 (2021)] but they indeed observed the 'magnon-phonon Fermi resonance' in this work as well (see Figure 3B of that paper). In this paper, however, they did not need to use the unique capabilities of the HZDR but rather carried the experiments out with a broadband tabletop THz source and instead of tuning the magnon frequency with a magnetic field they did so using temperature. Here as well they observed a resonant enhancement of the phonon amplitude when the condition $2 f_M(T) = f_{ph}$ is fulfilled. This is why I believe the author's claim that the HZDR facility is tantamount for this experiment and that the physics are novel is not true. I do agree with the authors that the THz facility at the HZDR is an incredibly valuable tool, yet I also think it is the wrong tool for the presented scientific case and no new insight is gained. Further, I still believe that the conclusion of a splitting phonon resonance hinges on the one data point at 3.5 T magnetic field. It is hard to judge that the splitting is real or just due to poor signal to noise (error bars are missing as well to assess this better). They further claim that "the magnon-phonon Fermi resonance the phonon spectral line (a) reduces in

amplitude, (b) broadens and (c) eventually splits at $H_{\text{ext}} = 3.5T$." I find this hard to see from figure 3a and can only stress again that the FFT line shape strongly depends on the FFT time windows. Also, I want to state here that this was my main concern about how the data points in Figure 3b were obtained (in the rebuttal Reviewer (2.6)), which the authors responded to in (Reviewer (2.7)). Another, major shortcoming of this work is the varying excitation density for different magnon frequencies (different magnetic fields). As observed by the authors this dramatically influences the observed amplitude of the magnon (Fig 3a). To address this, they "normalized" their data by the incident THz power (phonon) and the square root of THz power (magnon) (The THz power is nowhere to be found in the manuscript however). Unfortunately, this data treatment is not correct. For every data set (as function of magnetic field), they have to normalize the (phonon) Magnon amplitude by the (square of the) FFT amplitude at the magnon frequency or control the incident THz power in order to ensure equal magnon excitation densities. This has not been done. As I pointed out in my last review: "This problem can be circumvented by using either a tunable source or a broadband THz pulse." Sources capable to conduct this experiment in the "home" lab exist with respect of repetition, rate, frequency tunability and intensity. Therefore, I find their arguments not credible. Beyond these major issues there are other aspects where I disagree with the statements of the authors. Unfortunately, since my last review very little has changed, and I am not convinced that this manuscript should be published in Nature Communications. Yet, I believe with more revisions this paper could be suitable for a more specialized journal

Response (2.1): We respectfully disagree with the Reviewer and regret that the way the Reviewer responded to our counter-arguments does not facilitate any further discussion. We appreciate that Reviewers 1 and 3 do not agree with Reviewer 2 and share our point of view.

Reviewer #3 (Remarks to the Author):

Reviewer (3.1): The authors present a study of nonlinear magnon-phonon dynamics in antiferromagnetic CoF₂ for the case where the frequency of the magnon is nearly half the frequency of the phonon. By applying a magnetic field, the authors tune the frequency of the magnon into half-frequency resonance with the phonon mode, thereby strongly enhancing the energy transfer of the sum-frequency excitation process. The authors interpret this as a manifestation of Fermi resonance, supported by their measurements of change in spectral shape and intensity of the phonon.

In my opinion, this is an exciting study demonstrating a new mechanism of nonlinear magnon-phonon excitation, which deserves publication in Nature Communications.

The editor asked me to comment on the response by the authors to the reviews and in particular on point 2.5 of the response. In my opinion, the data is sufficient to support the claim of a magnon-phonon Fermi resonance. A direct 2-photon driving force (E^2) can be excluded due to the magnetic-field dependent spectral weight of

the photon (Fig. 2 in the response). This further excludes a 1-photon-1-magnon driving force (E^*l) as previously demonstrated by some of the authors, and leaves only the 2-magnon driving force (l^2) as a possibly source of the phonon excitation. Overall, I believe that the authors responded comprehensively to the comments by the reviewers and made appropriate changes to their manuscript.

Response (3.1): We appreciate the positive evaluation of our manuscript by Reviewer 3. Particularly, we are happy that the discussion on different terms for phonon excitation (E^2 vs L^2) which we provided in the previous rebuttal to Reviewer 2 is acknowledged, our data has been found to be sufficiently supporting the claim of a magnon-phonon Fermi resonance and our work deserves publication in Nature Communication. Indeed, the comparison of experimental data and numerical simulation in the strong coupling regime shows the best agreement for the L^2 term, emphasizing the validity of our claim.

Further questions and comments:

Reviewer (3.2): The 2-magnon-1-phonon scattering processes shown in Fig. 1c of the manuscript correspond to magnonic sum-frequency excitation of the phonon (left) and parametric down conversion of the phonon to two magnons (right), which the authors demonstrate here for the first time. What distinguishes the Fermi resonance the authors see from a simple combination of these two mechanisms? Figs. 4a, b suggests that the splitting of the phonon spectral weight occurs only at larger coupling strengths. Can the Fermi resonance therefore be seen as the strong-coupling regime of sum-frequency excitation and parametric down conversion?

Response (3.2): According to established terminology [Fermi, E. Zeitschrift fuer Physik 71(3-4), 250-259 (1931), Gornostyrev et al. Physical Review B 51(18), 12817-12820 (1995), Katsnelson et al. Physical Review B 66(9), 092303 (2002)], even a weak interaction between the phonon and the magnon modes can be seen as the Fermi resonance. In this case, the interaction can indeed be described as a combination of sum-frequency generation and down conversion. The Fermi resonance is well distinguished from any other mechanism in a strong coupling regime, when the resonance leads to dramatic changes in intensities and frequencies of the spectral lines. Hence, in response to the criticism we added a note to the main text stating that introduction of the term Fermi-resonance is essential for understanding the regime of strong coupling.

Reviewer (3.3): The authors point out in the discussion that nonlinear phononics is still a young field. A lot of work has been done in recent years for scattering processes related to that presented by the authors here however, and I think it would be worth connecting to some of them. For example, sum-frequency excitation in a 2-phonon-1-magnon scattering process has been reported recently [PRB 103, 094407 (2021)]. Furthermore, sum-frequency excitations for 2-phonon-1-phonon scattering processes have become increasingly popular in the field, including a recent demonstration by some of the authors, e.g. [PRL 131, 026902 (2023)], [PRB 97,

174302 (2018)], [PRB 102, 224301 (2020)], and [PRB, 97, 214304 (2018)]. In addition, the concept of enhancing a sum-frequency excitation process by tuning a quasiparticle frequency into a half-frequency resonance (here achieved by applying a magnetic field for the magnon) has been explored for a 2-phonon-1-phonon mechanism previously using optical cavities [PRResearch 3, L032046 (2021)]. Since the authors state in the introduction that the Fermi resonance is expected for any type of x^2*y coupling, this would make their findings generalizable to the strong-coupling regimes of the above coupling mechanisms as well.

Response (3.3): It is indeed true that Fermi resonance, in principle, can be generalized for any type of x^2*y coupling. We completely agree with Reviewer 3 that the field of nonlinear phononics has flourished in recent years, and the mentioned articles have greatly contributed to its advancement. In response to the criticism, we have changed the introduction of the paper aiming to present our findings as a demonstration of the strong-coupling regime.

Reviewer (3.4): Minor correction: Refs. [1] and [27] in the manuscript are the same.

Response (3.4): We thank the Reviewer for pointing out this oversight, and we have corrected these references.

Reviewer (3.5): To conclude, I recommend publication after addressing the above points. Response (3.5): We would like to thank Reviewer 3 for the positive evaluation of our manuscript. Additionally, we acknowledge the comments of the Reviewer, which stimulated us to better classify our mechanism of Fermi resonance among the many in the field of nonlinear phononics.

Reviewers' Comments:

Reviewer #3:

Remarks to the Author:

The authors have answered the questions I've raised and made appropriate changes in the revised manuscript. In particular, I appreciate the clarification of the Fermi resonance in the context of the sum-frequency and parametric downconversion processes in the strong-coupling regime.

I have no further remarks and recommend proceeding to publication with no further changes.